# In Search of Authenticity Biomarkers in Food Supplements Containing Sea Buckthorn: A Metabolomics Approach

**DOI:** 10.3390/foods12244493

**Published:** 2023-12-15

**Authors:** Ancuța Cristina Raclariu-Manolică, Carmen Socaciu

**Affiliations:** 1Stejarul Research Centre for Biological Sciences, National Institute of Research and Development for Biological Sciences, 610004 Piatra Neamț, Romania; ancuta.manolica@incdsb.ro; 2Faculty of Food Science and Technology, University of Agricultural Sciences and Veterinary Medicine Cluj Napoca, 400372 Cluj-Napoca, Romania; 3BIODIATECH—Research Center for Applied Biotechnology in Diagnosis and Molecular Therapy, 400478 Cluj-Napoca, Romania

**Keywords:** sea buckthorn, *Hippophae rhamnoides* L., commercial food supplements, authenticity biomarkers, metabolomics, UHPLC-QTOF-ESI^+^MS

## Abstract

Sea buckthorn (*Hippophae rhamnoides* L.) (SB) is increasingly consumed worldwide as a food and food supplement. The remarkable richness in biologically active phytochemicals (polyphenols, carotenoids, sterols, vitamins) is responsible for its purported nutritional and health-promoting effects. Despite the considerable interest and high market demand for SB-based supplements, a limited number of studies report on the authentication of such commercially available products. Herein, untargeted metabolomics based on ultra-high-performance liquid chromatography coupled with quadrupole-time of flight mass spectrometry (UHPLC-QTOF-ESI^+^MS) were able to compare the phytochemical fingerprint of leaves, berries, and various categories of SB-berry herbal supplements (teas, capsules, tablets, liquids). By untargeted metabolomics, a multivariate discrimination analysis and a univariate approach (*t*-test and ANOVA) showed some putative authentication biomarkers for berries, e.g., xylitol, violaxanthin, tryptophan, quinic acid, quercetin-3-rutinoside. Significant dominant molecules were found for leaves: luteolin-5-glucoside, arginine, isorhamnetin 3-rutinoside, serotonin, and tocopherol. The univariate analysis showed discriminations between the different classes of food supplements using similar algorithms. Finally, eight molecules were selected and considered significant putative authentication biomarkers. Further studies will be focused on quantitative evaluation.

## 1. Introduction

Sea buckthorn (SB, *Hippophae rhamnoides* L. or *Elaeagnus rhamnoides* (L.) A. Nelson, Figure 1) is a deciduous, dioecious thorny shrub belonging to the Elaeagnaceae family [1,2,3,4]. Native to regions of Europe and Asia, due to its high adaptability to extreme cold, drought, saline, and alkaline soils, sea buckthorn grows naturally or is cultivated nowadays on millions of hectares worldwide [3,4,5,6,7,8]. It is a versatile plant with a rich history and multiple ecological, economic, and therapeutical applications (Appendix A) [7,9,10,11]. The strong and complex root system with nitrogen-fixing nodules makes SB an optimal plant for soil and water conservation in eroded areas [12,13], and biodiversity protection [14]. In the food industry, SB is a valuable ingredient of food items such as jams, cheese, yogurt, fermented food, juices and other beverages, probiotic foods, or used as a food additive [10,15,16,17,18,19]. It can also supplement animal diets to improve the productivity and quality of final products [20,21,22,23].

The health-promoting properties of SB are attracting by far the most considerable attention from the research community, producers, and industry [11,24,25], becoming a common ingredient in a wide range of food supplements available on the markets [17]. Besides the large variability of composition due to its biological (genetic) strain, and geographical origin, many concerns are related to the authenticity of food supplements declared to contain SB components (berries or leaves). Contamination and adulteration of food supplements lead to variations in identity, purity, and expected benefits or therapeutic properties of the claimed botanical ingredient [26]. Therefore, finding new analytical approaches to ensure the quality and authenticity of food supplements is essential to minimize the potential risks related to their safe intake and to reach the expected nutritional and health-promoting effects [27,28].

All parts of sea buckthorn (berries, leaves, stems, shoots, bark, and roots) are used for their purported exceptional nutritional and health benefits [2,15,24,25,29,30]. The therapeutic activity of SB has been associated with its rich composition of nutritional and biologically active compounds (about 200) [9,25,31,32], particularly, high quantities of lipophilic antioxidants (e.g., carotenoids, tocopherols, phytosterols) and hydrophilic antioxidants (e.g., flavonoids, tannins, phenolic acids, ascorbic acid), among other constituents [11,32,33,34,35]. The small, orange-yellow colored berries, with a sour and astringent taste, are also rich and valuable ingredients in cosmeceuticals [36,37,38,39]. All anatomical parts of the berry (skin, flesh, endocarp, seed) have an impressive vitamin content, particularly vitamins C, A, and E [40,41,42], minerals [43,44], remarkable amounts of polyphenolic derivatives (mainly phenolic acids and flavonoids) [45,46,47], triterpenoids [48], carotenoids [35,49,50], fatty acids [34,44,51], and phytosterols (particularly β–sitosterol) [32,34,52,53]. Consumption of SB berries and derived preparations has been related to health-beneficial effects on the cardiovascular system (e.g., lipid metabolism, platelet aggregation, and inflammation) [54,55,56,57], glucose and lipid metabolism [58,59,60,61], and associated also with activities such as the immunomodulatory [62,63], antioxidant [64,65], antiviral [66,67], protective and curative effects in different pathologies [11,68,69,70,71]. The leaves and the new tender shoots have a similar chemical profile as berries but with significantly higher amounts of phenolic compounds [17,41,72,73,74,75], being a rich source of crude protein (on average 15%), crude fat, and macro- and microelements [33,42,76,77,78], being recommended in the production of new pharmaceutical or food ingredients and supplements [73,79,80]. The leaves have been reported to have anti-inflammatory [81,82], antioxidant [73,83], immunomodulatory [63], antimicrobial [84,85], anti-platelet and anticoagulant potential [86], as well as other health proprieties [87,88]. Other vegetative parts (e.g., stems, bark, roots), even if still underutilized, showed therapeutical potential [89,90,91], e.g., the root and stem have antioxidant and antimicrobial activity [92,93], while the bark has antimetastatic activity [94]. The by-products resulting from berry waste [95] and biomass (leaves and branches) [96] can be further valorized in the food industry, nutraceuticals, and cosmetics [97,98,99].

The phytochemical composition of SB is prone to variability under natural conditions that may be reflected in a high batch-to-batch variation of the chemical composition, critically altering the expected therapeutic effects. The chemical content varies among different parts of the SB plant [68,100], and in relation to the genotype, sampling location [101,102,103,104,105,106], gender (female and male) [89,107], developmental stages, post-harvesting procedures [89,108,109,110], and the extraction technology [108,109,111], all of these significantly influence the chemical content of the final preparation [108,109,111]. Furthermore, food supplements including SB (berries, leaves, lyophilized extracts) may often contain dozens of ingredients at different levels, making their quality control difficult since the standard analytical methods lack resolution within complex preparations [26,112].

Advanced analytical approaches, such as high throughput techniques (e.g., high-performance liquid chromatography-mass spectrometry (HPLC-MS), nuclear magnetic resonance (NMR) spectroscopy, or DNA-based methods) coupled with chemometric-guided approaches have recently attracted considerable attention in the fields of medicinal plants and derived herbal products [113,114,115,116,117,118,119]. The emerging field of plant metabolomics offers new strategies to determine the highly chemical variable profiles of plant materials [120]. Targeted and untargeted metabolomics strategies using different chromatographic techniques followed by a chemometric approach have been largely applied to document the metabolomic diversity of SB [75,121]. However, only a limited number of studies have reported on innovative analytical methodologies applied to authenticate SB commercially available products. Hurkova et al. [122] used direct analysis in real-time coupled with high-resolution mass spectrometry (DART-HRMS), ultra-high-performance liquid chromatography coupled with high-resolution mass spectrometry (UHPLC-HRMS), and high-performance liquid chromatography coupled with diode array detector (HPLC-DAD) to authenticate one SB food supplement (oil-based capsule) purchased at a hypermarket in the Czech Republic. Covaciu et al. [123] applied Raman spectroscopy, and gas-chromatography equipped with a flame ionization detector (GC-FID), combined with the supervised chemometric technique for oil differentiation, and found this suitable approach to detect possible adulteration of SB oil with sunflower oil. A multilayer perceptron-artificial neural network (MLP-ANN) was also tested in the same study [123]. Berghian-Grosan and Magdas [124] proposed a new, cost-effective approach for the control and authentication of edible oils, based on the rapid processing of Raman spectra using machine learning algorithms. In our previous studies, we applied ultra-high-performance liquid chromatography coupled with quadrupole-time of flight mass spectroscopy, and other techniques like Fourier Transform Infrared spectroscopy or UV-VIS spectroscopy for detecting and profiling phytochemicals in different food products, such as vegetable oils of different origins [125]. Despite the latest analytical advances, the authentication of botanical food supplements remains a major challenge due to the large diversity of contained ingredients that hinder the accuracy of analytical methods in identifying the targeted species and detecting the non-targeted species that may occur [126,127].

The objective of this study was to identify specific SB phytochemicals’ fingerprints in leaves and berries, as well as in various categories of commercialized food supplements (teas, tablets, capsules, syrups, or oils) to certify their presence, based on untargeted metabolomics procedure using ultra-high-performance liquid chromatography coupled with quadrupole-time of flight mass spectrometry (UHPLC-QTOF-ESI^+^MS). These data generated rapid and useful information on the presence and level of SB ingredients in different commercial supplements.

## 2. Materials and Methods

### 2.1. Samples Analysed

Twenty-three sea buckthorn-based commercial herbal supplements were randomly purchased from physical and online stores, including twelve herbal teas, three tablets, two capsules, four syrup/oils, and two dried berries (Table 1). Six genuine SB leaves (L1–L6) were kindly provided by our collaborators from “Anastasie Fatu” Botanical Garden, Iasi, Romania, and Agricultural Research and Development Station Secuieni (Secuieni, Neamt County, Romania). Voucher specimens were deposited at the National Institute of Research and Development for Biological Sciences, “Stejarul” Biological Research Centre (Piatra-Neamt, Romania), and are available on request.

### 2.2. Solvents, Reagents, and Analytical Standards

HPLC grade pure solvents (ethanol, acetonitrile, methanol, and tetrahydrofuran THF) were purchased from Merck (Darmstadt, Germany). Formic acid (99.99%) was purchased from Sigma-Aldrich (St. Louis, MO, USA). Deionized water was produced by a Milli-Q system (Millipore, Bedford, MA, USA).

### 2.3. Sample Preparation and Extraction of Phytochemicals

Each sample was finely grounded, and the powders (sieved particles smaller than 20 mesh (1.7 mm)) were subjected first to extraction in ethanol. The same quantity of 1 g from each powdered sample was suspended in 20 mL ethanol 50%, mixed for 15 min by vortex, and kept in an ultrasonic bath for 60 min at 50 °C. The suspension was kept for 24 h in the dark at room temperature, the extract was centrifuged at 12,500 rpm (4 °C) and the supernatant was collected and filtered through a 0.2 mm nylon filter. The procedure was repeated 2 times. To extract the lipophilic molecules after ethanol extraction, the pellet was mixed two times with 10 mL THF, sonicated in the ultrasonic bath for 3 × 20 min at 50 °C, left for 24 h in the refrigerator (2 °C), and then centrifuged at 12,500 rpm (4 °C). The THF extract (supernatant) was filtered through a polytetrafluoroethylene (PTFE) 0.25 mm filter. Both extracts (duplicated from each sample) were submitted to UHPLC-QTOF-ESI^+^MS analysis.

### 2.4. Untargeted Metabolomics Analysis Using UHPLC-QTOF-ESI^+^MS

The untargeted, metabolomic fingerprints of ethanolic extracts were performed using ultra-high-performance liquid chromatography coupled with electrospray ionization-quadrupole-time of flight-mass spectroscopy (UHPLC-QTOF-ESI^+^MS) on an UltiMate 3000 UHPLC system equipped with a quaternary pump Dionex delivery system (Thermo Fisher Scientific Inc., Waltham, MA, USA), and mass spectroscopy (MS) detection by a QqTOF MaXis Impact (Bruker Daltonics GmbH, Bremen, Germany). The metabolites were separated using a 5 μm Kinetex column (Phenomenex Inc, Torrance, USA) (2.1 × 150 mm) at 25 °C. The flow rate was set at 0.8 mL·min^−1^ and the volume of each injected extract was 10 μL. The mobile phase consisted of 0.1% formic acid in water (A) and 0.1% formic acid in acetonitrile (B). The gradient was 20–40% B (0–5 min), 40–60% B (5–8 min), 60–70% B (8–10 min), 70–20% B (10–16 min), and 20% B isocratic until 24 min. Several quality control (QC) samples obtained from each extract group were used to optimize the separations. The chromatograms were processed using Chromeleon software (Dionex, Thermo Fisher Scientific Inc., Waltham, MA, USA). The MS parameters were ionization mode positive ESI+, calibrated with sodium formate, capillary voltage 3500 V, nebulizing gas pressure of 2.8 bar, drying gas flow 12 L/min, drying temperature 300 °C. The resolution of triple-quadrupole-TOF was 30,000 at *m*/*z* = 922. The control of the instrument and the data processing were done using the specific softwares TofControl 3.2, HyStar 3.2, and Data Analysis 4.2 (Bruker Daltonics GmbH, Bremen, Germany).

#### Data Processing and Statistical Analysis

The Bruker software Compass Data Analysis 4.2 (Bruker Daltonics, GmbH, Bremen, Germany) was used to process the MS spectra of each component separated by chromatography. The base peak chromatograms (BPC) were obtained from the total ion chromatogram and by the algorithm Find Molecular Features (FMF), a bucket matrix was generated, including the mass-to-charge ratio (*m*/*z*) value for [M + 1]^+^ precursor molecules, the retention time, the peak intensity, and the signal/noise (S/N) ratio. The initial number of separated molecules (*m*/*z* values) was around 550. The alignment of common molecules (with the same *m*/*z* value) was done by the online software (www.bioinformatica.isa.cnr.it/NEAPOLIS (accessed on 19 September 2023)). A second matrix of the common molecules found in more than 60% of samples was obtained, having S/N values over 2 and peak intensities over 10,000 units. The resulting data matrix included a few 98 *m*/*z* values versus peak intensity and was submitted for statistical analysis in the Metaboanalyst v5.0 online software for multivariate and univariate (one-way ANOVA) analysis.

The statistical algorithms used to reflect the discrimination between the different sample groups were the partial least square discriminant analysis (PLSDA), the variable importance in the projection (VIP) scores, and the correlation heatmaps. The biomarker analysis included the receiver operating characteristic (ROC) curves and area values under ROC curves (AUC) values which evaluated the sensibility and selectivity of the potential biomarkers. According to the statistical analysis, the candidate molecules for authenticity to be considered putative biomarkers were selected and identified, using the specialized database FoodDB (https://foodb.ca/, accessed on 25 September 2023). The multivariate metabolomic analysis was used to compare the leaves (L1-L6) with dried berries (B1–B2) to find the most relevant molecules that may discriminate the phytochemicals specific to leaves versus berries. The data from the univariate one-way ANOVA analysis was applied to find out the discriminations between the different classes of molecules found in the food supplement samples that claimed the presence of SB berries in the composition. In both cases (the *t*-test and significance of differences (*p*-values and post-hoc Fisher LSD) were calculated.

## 3. Results

### 3.1. UHPLC-QTOF-ESI^+^MS Untargeted Analysis

The untargeted analysis was performed using multivariate and univariate analysis, and showed possible discriminations between the supplements (groups B, S, C, Tb, and T) which claimed to contain SB berries as such, or extracts as ingredients in their composition, at different levels. No clear indication of the concentration or the percentage of SB herbal components was provided by the product labels. Such analysis aimed to identify some specific phytochemicals that may indicate at least qualitatively the presence of berries in FS.

For the metabolomic analysis, based on the MS data (matrix including *m*/*z* values versus peak intensity) 98 molecules were identified according to the described procedure in Section 2.4. The experimental *m*/*z* values were compared with the average *m*/*z* values from FooDB (https://foodb.ca/, accessed on 25 September 2023). The list of identified phytochemicals is presented in Appendix A. Only molecules having the accuracy of (theoretical—experimental) *m*/*z* values below 20 ppm were considered. For each molecule, the FooDB code was mentioned.

#### 3.1.1. Multivariate Analysis

##### PLSDA, Fold Change and *p*-Values

Figure 2 presents the PLSDA score plot which reflects the discrimination between the SB leaves (L) versus berry (B) composition according to PLSDA analysis (co-variance of 67.3%). Despite the small number of samples, the cross-validation algorithm showed the highest accuracy, with high R2 values and a significant Q2 value (>0.93) for the third component, confirming the good predictability of this model (Appendix A). The VIP score graph (ranging from 1.2–1.5 values), derived from PLSDA analysis, was also done including the ranking of the molecules that may explain the discrimination between groups L and B. The VIP scores identified the molecules responsible for the discrimination, either at superior levels in the B group (marked in red) or inferior in the L group (marked in green).

The Fold change (FC) and the log2(FC) values, according to the Volcano plot algorithm (shown as Appendix A) and the PLSDA/VIP analysis, were useful in identifying the molecules with increased or decreased levels when comparing the group L with group B.

Table 2 describes the FC values, log2(FC) combined with the *p*-values according to the *t*-test.

These parameters and the sign of the log2(FC) show the top of 20 molecules from quercetin-3 rutinoside to glucose as being more dominant in berries (positive log2FC values) and phytoene to ferulic acid being more dominant in leaves (negative log2FC values). Considering the lowest *p*-values (<0.0001), in each case, for berries, the putative biomarkers to be considered were xylitol, violaxanthin, folic acid, tryptophan, quinic acid, quercetin 3 rutinoside. For leaves, significant dominant molecules were luteolin 5-glucoside, arginine, isorhamnetin 3-rutinoside, serotonin, and tocopherol. This data was compared also with complementary information given by the heatmap.

##### Heatmap Plot and Biomarker Analysis

The heatmap plot (Figure 3) illustrates the different clustering of the groups L and B as well the relationships between molecules (increase or decrease in the groups L and B).

This represents complementary information and illustrates by colors the levels of the molecules in the B group (PC5, 6, 14) compared to group L (ACM 1, 2, 4, 5, 6, 7). We can distinguish higher levels of quinic and feruloyl quinic acid and xylitol, violaxanthin, folic acid, tryptophan, and cis retinal to be also of interest for discrimination between leaves and berries, with significant increases in berries. Considering that all investigated supplements claimed to contain SB berries or extracts of SB berries, the next studies were focused on these molecules.

According to the biomarker analysis, the highest AUC values (>0.9) for the molecules to be considered putative biomarkers for berries were found also to be xylitol, violaxanthin, folic acid, tryptophan, quercetin-3-rutinoside, and quinic acid.

#### 3.1.2. Univariate One-Way ANOVA Analysis to Evaluate the Discrimination between the Different Classes of Food Supplements

##### sPLSDA and Heatmap

The different supplements (teas, tablets, capsules, syrups/oils) were considered for the one-way ANOVA analysis. The dried berries (group B) were unified in this case with the liquid samples resulting in a group BS, the same for the groups C and Tb, named CTb. Therefore, we compared the teas (group T) with groups BS and CTb. Figure 4A shows the sPLSDA score plot and Figure 4B the loadings plot showing the top 15 molecules responsible for the discrimination between the 3 groups (BS, CTb, and T). The relative levels are presented on the right side (red-high; blue-low).

According to Figure 4A, a good discrimination between teas (blue region), CTb group (green region), and BS group (pink region) was identified. The loadings plot shows variations among the molecules identified as putative biomarkers for berries: higher levels in the BS group for miristoylcarnitine, gallocatechin, cis-retinal, riboflavin, violaxanthin, quinic acid, quercetin-3-rutinoside. This data confirms that some of these molecules can be considered biomarkers for the berry’s extracts (syrups or SB oil) by multivariate analysis. Comparatively, the levels of these molecules in groups T or CTb were inferior. Figure 5 illustrates the heatmap data, as complementary information to show the presence of SB berries in groups T and CTb.

Significant discrimination was also illustrated here, between the groups BS, CTb, and T. In the BS group, we identified higher levels of violaxanthin, tryptophan, carotene, catechin, feruloylquinic acid, and neoglucobrassicin while in the CTb group, we identified higher levels of glucose (additive), zeaxanthin, and hydroxytryptophan (possibly as additives). The group of teas (T) showed especially higher levels of serotonin, gallic acid, kaempferol 3-rhamnoside, and some unidentified molecules, from the plant mixtures used in the formulations.

Since this analysis was not satisfactory enough to find the lower levels of SB berries present in teas and CTb groups, we also evaluated some specific molecules.

### 3.2. Evaluation of the Selected Putative Biomarkers

Based on the data cumulated from the multivariate and univariate analysis, several molecules were selected as putative biomarkers for SB berry phytochemicals in such a diverse cohort of botanical products, as an indication of authenticity. Figure 6 represents the levels of eight molecules (xylitol, quinic acid, tryptophan, folic acid, quercetin-7-glucoside, violaxanthin, quercetin-3-rutinoside, quercetin-3,7-diglucoside), previously selected by multivariate and univariate analysis. The levels were evaluated based on their peak intensities in the UHPLC-MS analysis.

The comparative evaluation shows that the variability of composition is maintained but is closer to a more adequate consideration of authenticity. Capsules C1 and C2 showed significantly lower levels, which may be explained by higher percentages of excipients, compared to tablets (Tb1–Tb3) which showed a more stable composition. The tea composition was variable, except for the level of tryptophan which proved to be a major component compared to other molecules. Further quantitative evaluation of such molecules will bring more valuable information for a selection of representative SB biomarkers in herbal supplements.

## 4. Discussion

Applying innovative techniques to advance food supplements authentication is strongly advocated today [27,113,114,116,125,128].

Considering the high market demand for SB-based products, its phytochemistry and pharmacognosy have stimulated considerable interest, but a limited number of studies on the quality and authenticity of commercially available food supplements are reported [122,123,124]. However, significant progress has been offered in the last years by the methodological approaches that combine advanced analytics with multivariate statistics, particularly for SB berries [34,35,37,46,52,53,75].

Metabolomics is an accurate, robust, and time-efficient analytical approach for the authentication of different molecules in complex botanical products. The emerging field of plant metabolomics offers new ways to determine the profiles of plant bioactive compounds as such, which are highly variable under the influence of various factors (genetic, environmental, processing technology), and, on top of this, allow their measurement in complex commercial botanical products, such as food supplements. The untargeted metabolomics can offer improved fingerprints and resolution of the authentication process of botanical-based foods and food supplements. Comprehensive reviews on integrated analytical approaches and chemometric-guided approaches for profiling and authenticating botanical materials applied to the identification of botanical bioactive compounds and adulteration management were previously published [113,129,130].

Authentication is challenging when plant material is powdered or extracted in different solvents, as well as for mixtures consisting of multiple plant species. Moreover, tracing bioactive phytochemicals claimed on the labels of botanical food supplements is complicated by the natural variability of the starting raw material which often results in a significant variation in the composition of the final product. Nevertheless, the deliberate replacement of bioactive ingredients, their dilution, or the addition of lower-cost ingredients, is a significant ongoing problem in this sector. Nowadays, the accurate recognition of phytochemicals within a complex mixture and the identification of specific bioactive compounds from plant components (leaves, berries) requires the use of orthogonal, fused, and specific analyses, including multivariate, univariate analysis coupled with chemometrics [113,130].

Our study aimed to demonstrate the added value of the metabolomic approach for finding key phytochemicals originating from sea buckthorn (leaves or berries) and different food supplements including teas, capsules, tablets, syrups, and oils.

Using UHPLC-QTOF-ESI^+^MS untargeted (multivariate and univariate) analysis in conjunction with multivariate analysis, using PLSDA score and loadings plots, heatmap, the Fold change, and *t*-test, we found that the putative authentication biomarkers (*p* values < 0.0001) of SB berries are xylitol, violaxanthin, folic acid, tryptophan, quinic acid, quercetin-3-rutinoside. For leaves, luteolin-5-glucoside, arginine, isorhamnetin 3-rutinoside, serotonin, and tocopherol were found to be significant dominant molecules. The univariate analysis aimed to discriminate between the different classes of food supplements (BS, CTb, and T) using similar algorithms. The sPLSDA plots showed good discrimination between teas (T), CTb, and BS groups and reflected putative biomarkers for berries (higher levels in the BS group for miristoylcarnitine, gallocatechin, cis-retinal, riboflavin, violaxanthin, quinic acid, quercetin-3-rutinoside). The heatmap illustrated the presence of SB berries in groups T and CTb but at lower levels. In the BS group, we identified higher levels of violaxanthin, tryptophan, carotene, catechin, feruloylquinic acid, while in the CTb group, higher levels of glucose (additive), zeaxanthin, and hydroxytryptophan (possible additives). The group of teas (T) showed especially higher levels of serotonin, gallic acid, and kaempferol 3-rhamnoside and some unidentified molecules, from the plant mixtures used in the formulations.

Since this analysis was not satisfactory enough regarding the lower levels of these molecules in T and CTb groups, we also considered a semiquantitative evaluation of the eight selected molecules (xylitol, quinic acid, tryptophan, folic acid, quercetin-7-glucoside, violaxanthin, quercetin-3-rutinoside, quercetin-3,7-diglucoside) as SB berry biomarkers, according to their peak intensities in the UHPLC-QTOF-ESI^+^MS untargeted analysis. The comparative evaluation shows that the variability of composition is maintained but is closer to a more adequate consideration of authenticity. Capsules C1 and C2 showed significantly lower levels, explained by higher percentages of excipients, while tablets (Tb1-Tb3) showed a more stable composition. The teas’ composition was variable, except for the level of tryptophan, found as a major component compared to other molecules. These molecules can represent a starting point for a further quantitative evaluation of some key molecules selected here as putative biomarkers of the presence and level of SB berry components in botanical food supplements.

A single plant species produces far more metabolites than those produced by most other organisms [131,132], and, so far, no stand-alone analytical approach has been able to untangle this diversity [127,131]. Additionally, complex plant-based food supplements contain numerous plant ingredients, or mixtures of plant and vitamins or mineral ingredients, among others, hindering, even more, the resolution of analytical methods in identifying the targeted species and detecting the non-targeted species that may occur [126,127]. Moreover, there is a large body of evidence that unexpected contaminants and/or adulterants are often present in such herbal matrices [26]. Therefore, orthogonal testing approaches that include multiple complementary analytical methods are recommended to comprehensively elucidate the ingredients and chemical content of herbal products [26,120,133,134].

## 5. Conclusions

The authentication of botanical food supplements based only on specific bioactive plant phytochemicals remains a major challenge despite the latest advances in analytical technologies. Even the more advanced analytical methods are not powerful enough to identify qualitatively, and especially quantitatively, the biomarkers of authenticity for a specific ingredient, for instance, sea buckthorn. In this study, untargeted metabolomics based on UHPLC-QTOF-ESI^+^MS was performed for the identification of the phytochemical profiling of SB food supplements. This study presented three steps of analytical flow, from preliminary spectrometric analysis to multivariate and univariate metabolomic fingerprinting, finalized by a semiquantitative evaluation based on the MS peak intensities of selected phytochemical biomarkers, useful to authenticate food supplements declared to contain sea buckthorn components (leaves or berries). Finally, there is an urgent need to apply orthogonal advanced analytical approaches to fully untangle the huge ingredient and chemical diversity of commercial botanical products.

## Figures and Tables

**Figure 1 foods-12-04493-f001:**
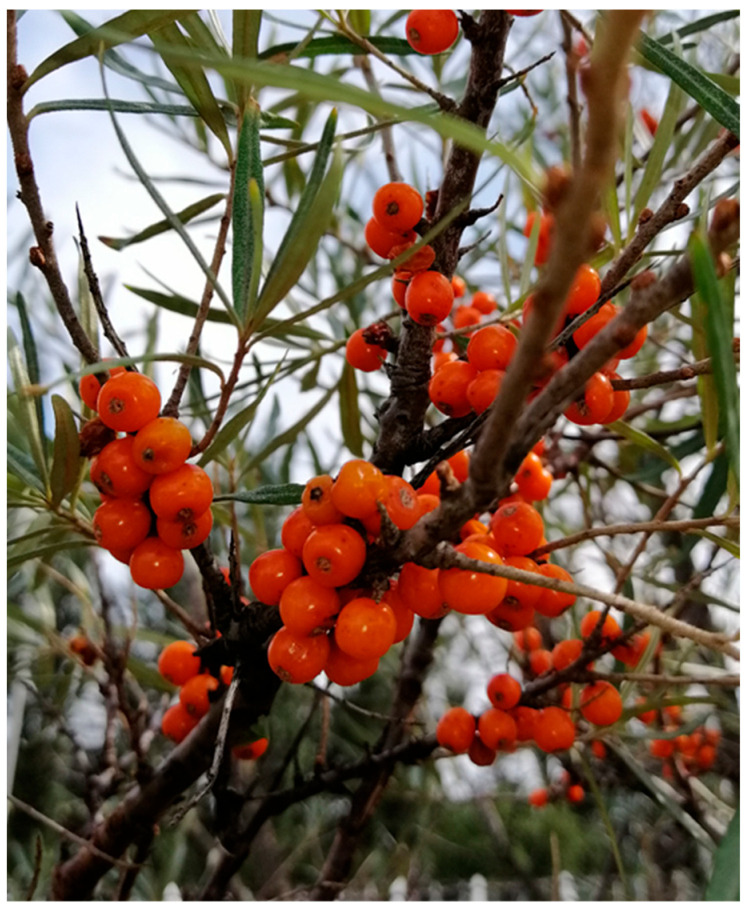
Sea buckthorn (*Hippophae rhamnoides* L. or *Elaeagnus rhamnoides* (L.) A. Nelson). Branch with red-orange ripe berries, thorns, and leaves (Photos taken at the Agricultural Research and Development Station (SCDA) Secuieni, Neamt County, Romania by A.C. Raclariu-Manolică).

**Figure 2 foods-12-04493-f002:**
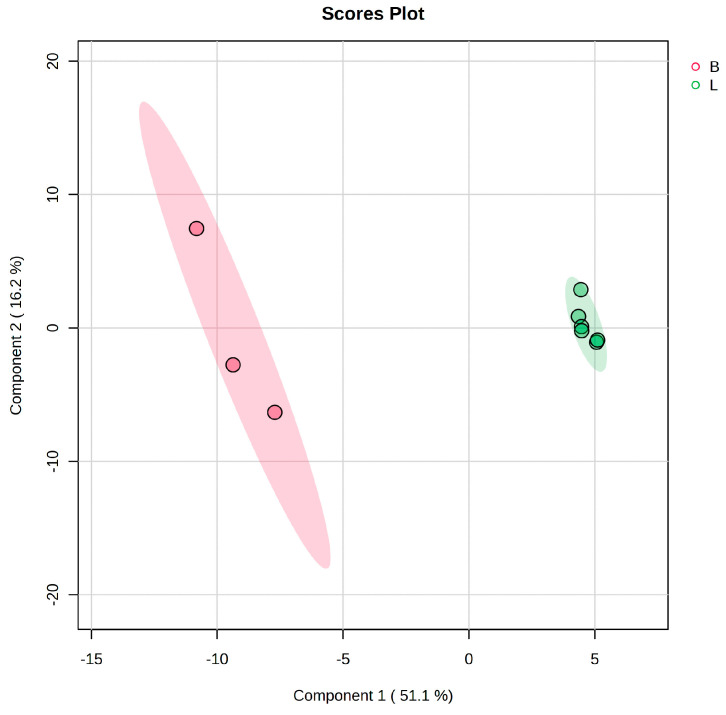
PLSDA score plot showing the discrimination between the groups leaves (code L) and berries (code B).

**Figure 3 foods-12-04493-f003:**
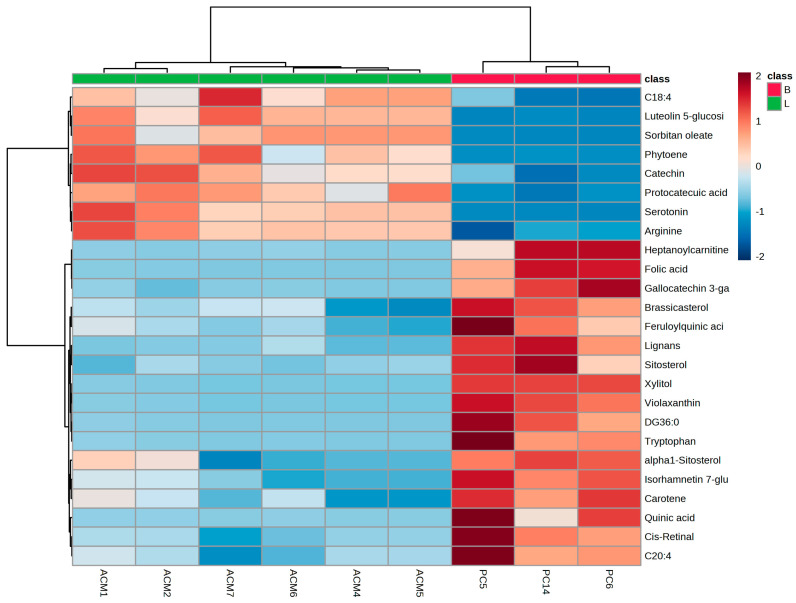
The heatmap showing the clusters of groups of leaves (ACM1, 2, 4, 5, 7) and berries (PC5, PC6, PC14) considering the mean values for the first 25 molecules selected as most relevant for discrimination.

**Figure 4 foods-12-04493-f004:**
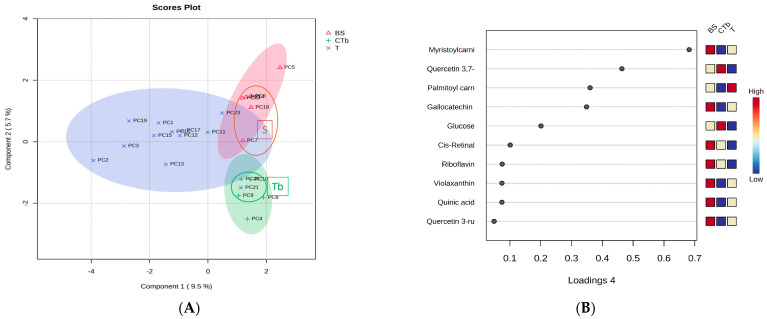
(**A**). sPLSDA score plot shows the discrimination between the groups BS, CTb, and T. (**B**). The loadings plot of the top 15 molecules responsible for the discrimination between the 3 groups (BS, CTb, and T). The relative levels are presented on the right side (red-high; blue-low).

**Figure 5 foods-12-04493-f005:**
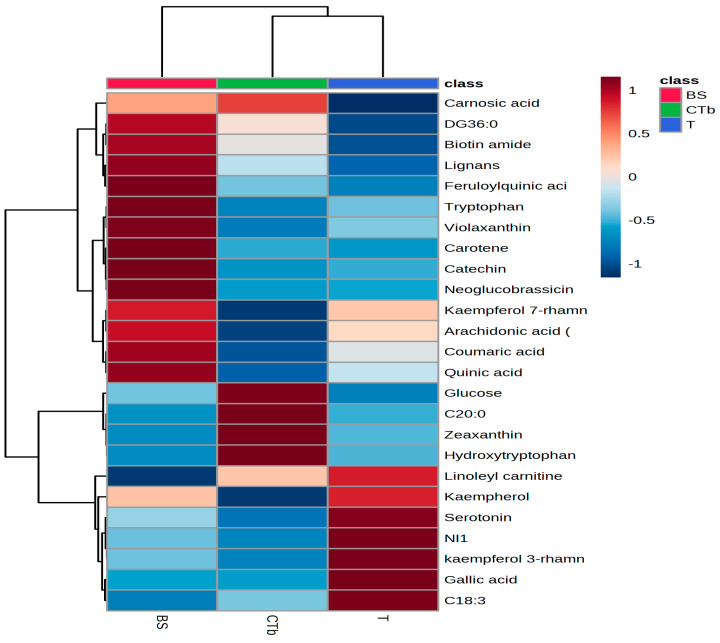
The heatmap for the groups BS (berries, syrup/liquids), CTb (capsules, tablets), and T (teas), considering the mean values for the first 25 molecules selected as most relevant for the discrimination among these groups.

**Figure 6 foods-12-04493-f006:**
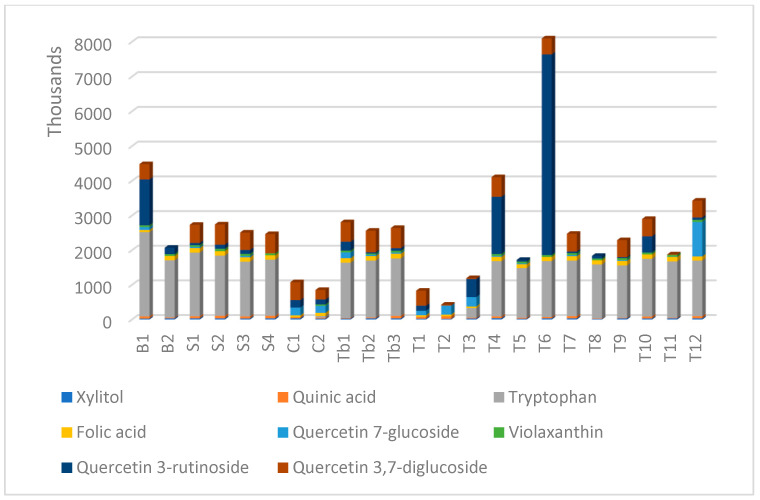
Semiquantitative analysis of phytochemicals specific to SB berries, found in the different supplements (T—teas; Tb—tablets; C—capsules; S—syrups/oils; B—Dried Berries): the levels of different molecules (xylitol, quinic acid, tryptophan, folic acid, quercetin-7-glucoside, violaxanthin, quercetin-3-rutinoside, quercetin-3,7-diglucoside) according to their peak intensities in the UHPLC-QTOF-ESI^+^MS untargeted analysis.

**Table 1 foods-12-04493-t001:** Categories of herbal formulations used for scientific analysis, and their collection and analysis codes. Abbreviations used: T—Tea; Tb—Tablet; C—capsule; S—liquid supplement; B—Berry; L—Leaves.

Type of Formulation	ID Collection Code/ID Analysis Code
Herbal tea (T)	PC1/T1
	PC2/T2
	PC3/T3
	PC11/T4
	PC12/T5
	PC13/T6
	PC15/T7
	PC16/T8
	PC17/T9
	PC19/T10
	PC21/T11
	PC23/T12
Tablet (Tb)	PC9/Tb1
	PC10/Tb2
	PC20/Tb3
Capsule (C)	PC4/C1
	PC8/C2
Syrup/Oil (S)	PC6/S1 (oil)
	PC7/S2 (hydroalcoholic extract)
	PC18/S3 (emulsion)
	PC22/S4 (syrup)
Dried Berry (B)	PC5/B1
	PC14/B2
Leaves (L)	ACM1/L1
	ACM2/L2
	ACM4/L3
	ACM5/L4
	ACM6/L5
	ACM7/L6

**Table 2 foods-12-04493-t002:** Fold change (FC), log2(FC) values, and *p*-values according to PLSDA analysis and *t*-test. The significance of variation between groups B and L (B > L or B < L) is presented. In Bold are represented the most significant ones.

B > L	FC	log2(FC)	*p*-Value	L > B	FC	log2(FC)	*p*-Value
**Quercetin-3-** **rutinoside**	**69.666**	**6.122**	**0.0100**	Phytoene	0.017	−5.889	**0.0012**
Stigmasterol	44.887	5.488	**0.0103**	Acetylspermidine	0.023	−5.442	**0.0042**
Hydroxy tryptophan	26.948	4.752	**0.0167**	DiGlyceride 30:2	0.033	−4.906	**0.0182**
Biotin amide	26.909	4.75	**0.0031**	**Tocopherol**	**0.035**	**−4.834**	**0.0070**
Naringin	21.41	4.42	**0.0420**	Caffeic acid	0.044	−4.512	**0.0450**
Lauroyl carnitine	19.186	4.262	**0.0046**	**Serotonin**	**0.074**	**−3.75**	**0.0001**
**Quinic acid**	**17.721**	**4.147**	**0.0025**	Gallic acid	0.079	−3.658	**0.0460**
Fatty acid C20:0	15.023	3.909	**0.0450**	Sorbitan oleate	0.107	−3.23	**0.0001**
Fatty acid C12:0	13.965	3.804	**0.0470**	**Luteolin-5-glucoside**	**0.129**	**−2.959**	**0.0000**
**Folic acid**	**13.405**	**3.745**	**0.0001**	Hydroxyglutamine	0.141	−2.826	**0.0470**
Arabinose	13.013	3.702	**0.0053**	Kaempferol 3-rhamnoside, 7-glucoside	0.149	−2.744	**0.0076**
Heptanoyl carnitine	10.675	3.416	**0.0017**	Fatty acid C18:4	0.15	−2.739	**0.0018**
Quercetin-7- glucoside	9.976	3.318	**0.0470**	Fatty acid C20:2	0.156	−2.678	**0.0039**
DG36:0	9.654	3.271	**0.0480**	Glucuronic acid	0.17	−2.553	**0.0068**
**Tryptophan**	**9.470**	**3.243**	**0.0003**	Fatty acid C18:3	0.277	−1.852	**0.0470**
Glucitol	9.202	3.202	**0.0040**	**Arginine**	**0.283**	**−1.819**	**0.0002**
**Xylitol**	**8.836**	**3.144**	**0.0000**	**Isorhamnetin 3-** **rutinoside**	**0.292**	**−1.776**	**0.0002**
**Violaxanthin**	**8.11**	**3.02**	**0.0000**	Luteolin	0.312	−1.679	**0.0490**
Vanillic acid	6.187	2.629	**0.0164**	Myristoylcarnitine	0.331	−1.596	**0.0041**
Glucose	5.89	2.558	**0.0154**	Ferulic acid	0.335	−1.578	**0.0070**

## Data Availability

Data is contained within the article or Appendix A.

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
