# Peer review of "In Search of Authenticity Biomarkers in Food Supplements Containing Sea Buckthorn: A Metabolomics Approach"

_foods, 2023, doi:10.3390/foods12244493_

Round 1
Reviewer 1 Report
Comments and Suggestions for Authors
Review Manuscript ID: foods-2739002
Title:
In search of authenticity biomarkers from diverse food supplements containing sea buckthorn: a metabolomics approach
This title does not present the acquired data presented in results section.
Abstract:
OK
Key words:
OK
Introduction:
The bioactive ingredients of the study material is well documented, however, no background is provided regarding the analytical work done especially from metabolomics point of view. Adding this aspect to the introduction is recommended. Additionally there is a mention of adulteration and QC issues.
Materials and methods:
b= berries, l=leaves, s=oil, tb=tablet, t=tea
L60-64. Figure 1 is taking a significant space. Please check the journal’s policy before including in finally submitted manuscript.
L142-145. Figure 2 is taking a significant space. Please check the journal’s policy before including in finally submitted manuscript.
L156-167. 2.3. Sample preparation and extraction of phytochemicals
Add details of material grinding (mash size)
Include temperatures at which centrifugation process was performed and referegration was done.
L188. Injection volume 10 mL is mistyped. Please add correct injection volume.
Why foodb.ca was preferred over for example Chemispider and HMDB as external search engine?
Results:
Overall results are not presented in a way that clearly showcase the power of accurate mass high resolution data and the post acquisition of statistical evaluations.
L230-292. 3.1. UV-VIS spectral fingerprints in ethanol 50% and THF extracts.
Over the results are inconclusive due to the fact that no QA/QC evaluations are benchmarked to validate the obtained data and thus the variations shown in bar graphs and commented in results can not be extrapolated to make semi quantitative conclusions even at semiquant level. I recommend to remove these results and associated experimental portions because accurate mass high resolution data should cover several aspects regarding extract and compound identify.
L295-301. These statements are not results.
L293-307. 3.2. UHPLC-QTOF-ESI+MS untargeted analysis
This section is inconclusive and needs further information regarding the analytical findings including a list of identified phytochemicals, mass accuracies, comparative intensities and putative identities. The outcomes of the searches on food bank search.
Comparative results from foodb.ca and / or Chemispider and HMDB as external search engines is recommended due to robustness of later.
L310-397. 3.2.1 Multivariate analysis
At metabolome the crude extracts of leaves fruits always differ due to distinct metabolic events in these plant organs but how the clustering of all study materials would look like should be presented as a separate scores plot followed by discriminant analyses for chemical comparisons.
The absence of standards (all are common) injected for confirmation is a major limitation of this study.
Figure 6A, codes do not match the ones listed in Table 1. Assuming CTa = tablet, BS = Barries and T = tea, t1 = 9.5% and t2 = 5.7% do not represent clear discrimination although the materials are different preparations. This aspect should be highlighted. L382-383. “Comparatively, the levels of these molecules in groups T or CTb 382 were inferior” does not clearly define the results.
Figure 7 is confusing and plausible conclusions can not be easily drawn.
L399- 3.3. Targeted analysis.
Targeted analysis essentially include quantitative analyses. The presented results are semi quantitative, figure 8.
Discussion:
L423-443. Discusses the QC aspects. However the study structure does not present these experiments and thus these discussions are outside the scope of this stud.
No comparative literature review is presented that can show the importance and novelty of this study.
Conclusion:
The conclusion highlights the QC aspect of the study.
Supplementary Material
Some data related to QC should be added to the main study to add value.
Author Contributions
OK
Funding:
Not evaluatged
Data availability:
Not evaluated
Acknowledgements:
Ok
Conflicts of Interest:
Ok
References:
The references are not focused on metabolomics aspects of the study.
Additional references should be added to enhance discussion section.
Comments on the Quality of English LanguageEnglish language should be reviewed to improve this manuscript.
Author Response
The answers are attached

Reviewer 2 Report
Comments and Suggestions for Authors
The manuscript from Manolica et al. aims to identify specific phytochemical fingerprints in sea buckthorn (fruits, leaves) and derived products by UV-VIS spectroscopy and untargeted LC-MS analysis.
The food biomarker is very challenging, due to the fact that many of the phytochemicals/molecules are present in several different fruits and finding the specific ones for a specific kind of food is not easy. I believe this point should be addressed and also how would you test that the patters/molecules you have found are only related to SB.
Regarding the choice of the samples,
1) why fresh berries are not included in the sample set? Are you considering the dried berries as an example for the real berry or a processed food?
2) the leaves were dried or fresh? Were they from different SB species?
3) Why not including more SB berries (more than 2) in the sample set, since you utilized the discrimination between leaves and berries to generate a pattern of possible molecules characteristic for the SB? The analysis is not balanced (more leaves samples than berries)
4) The composition of the herbal tea was not mentioned, but has stated it may contain different kind of herbs, which could also potentially contain same compounds, if other berries or fruits were present. I think it should be mentioned what kind of tea were used.
5) I think it would have been better to test the findings from the PLSDA analysis leaves vs berries, berries were not supposed to be present in the second analysis, because you could have used this sample set (only products) to test the model (as independent sample set)
Furthermore, I have more detailed comments:
1) NMR and UV are spectroscopy techniques not spectrometry. Please correct throughout the manuscript.
2) Lines 24:25. I would say that you used both a multivariate (PLSDA is a whole analysis, not only the plots) and a univariate approach(T-test, and ANOVA)
3) Line 124. You did not fused the data, you simply used two different analytical techniques and compared.
4) Figure 2. It would be better if you would show them as you group them in Table 1. It is not easy to see what is what in the commercial product. Maybe you could add the info from Table 1 to the figure, then you would not need the table any longer or just have a more detailed caption for Figure 1.
5) Lines 157:164. What the 3 times stands for? Did you repeat the extraction 3 times per samples?
6) Line 188. I believe you meant 10uL and not ml
7) Line 192. What do you mean with calibrating the separations?
8) Line 201-211. The original dataset after peak integration were 550, but the only one having a S/N >2 and present in at least 60% of the samples were 98? Why did you not look also into specific molecules present only in specific group of samples?
9) The PLSDA analysis is not well described. There are missing details about cross validation (what kind and the subset size), VIP level selected. What kind of T-test was used?
10) Line 320. I guess you meant green
11) Line 325. It would be very good to show the Volcano plot (which is not an algorithm)
12) Line 362. It is not only univariate because you have PLSDA analysis below.
13) Line 366. Did you mean PLSDA and not ANOVA? You did not mentioned the goodness of the PLSDA model now that you have 3 groups. Where is the ANOVA analysis? And you mentioned in the methods section about p-values and post-hoc Fisher analysis from the ANOVA.
14) Line 399. I would not called a targeted evaluation of the biomarker. It is a summary of the main findings from the univariate/multivariate analysis in a bar plot.
15) Figure S1. It is somehow intuitive but it could be improved. Firstly you could you use the same color shades for the 2 main specimens: leaves vs fruits. And the blue box seems a bit repetitive since you have a scheme above.
16) Figure S2(A). What is OD? And in the caption you mentioned kaempherol, you mean rutin? Are these UV-VIS spectra similar for the different class of compounds? Because you mentions carotenoids, chlorophyll derivatives and phenolic acids as all class but in the spectra you mention specific molecules. Can you discriminate between different molecules within the same class? Otherwise I would just leave it as you also mentioned in the main text as a main class and keep the figure more general, also because you do not mention any specific molecules for the spectrophotometric analysis.
Author Response
The answers were attached

Round 2
Reviewer 1 Report
Comments and Suggestions for Authors
the answers of the questions/ comments provided by the corresponding author are ok.
Comments on the Quality of English Languageminor editing
Author Response
Point-by-point answers are attached..

Reviewer 2 Report
Comments and Suggestions for Authors
Some minor things should still be fixed as follows:
1) Including having the volcano plot on the supplementary material (since it is mentioned in the main text it would be a good idea to show it in the supplementary)
2) According to me, it is still confusing showing a PLSDA plot when the analysis specified is ANOVA. I tried to follow the link they provided, and it did not work. On the more general one it does not show or exactly say how the 2 different analyses are performed. PLSDA is a supervised classification model based on a multivariate approach, while the ANOVA is a univariate approach. There is something called ASCA, which is a fusion of ANOVA and PCA, but I am not sure what you used. Furthermore, in the main text, you stated that it is a sPLSDA, which is also a different PLSDA analysis which is also not mentioned in the method section.
These are my main comments, thank you.
Comments on the Quality of English Language
ok
Author Response
Answers are attached, thank you !!
